# Intravesicular Genomic DNA Enriched by Size Exclusion Chromatography Can Enhance Lung Cancer Oncogene Mutation Detection Sensitivity

**DOI:** 10.3390/ijms232416052

**Published:** 2022-12-16

**Authors:** Rebekka Van Hoof, Sarah Deville, Karen Hollanders, Pascale Berckmans, Patrick Wagner, Jef Hooyberghs, Inge Nelissen

**Affiliations:** 1Health Unit, Flemish Institute for Technological Research (VITO), 2400 Mol, Belgium; 2Laboratory for Soft Matter and Biophysics, KU Leuven, 3000 Leuven, Belgium; 3Theoretical Physics, Hasselt University, 3590 Diepenbeek, Belgium; 4Biomedical Research Institute, Hasselt University, 3590 Diepenbeek, Belgium; 5Data Science Hub, Flemish Institute for Technological Research (VITO), 2400 Mol, Belgium

**Keywords:** extracellular vesicles, non-small cell lung cancer, *EGFR* T790M, genotyping, size exclusion chromatography, intravesicular DNA, digital droplet PCR

## Abstract

Extracellular vesicles (EVs) are cell-derived structures surrounded by a lipid bilayer that carry RNA and DNA as potential templates for molecular diagnostics, e.g., in cancer genotyping. While it has been established that DNA templates appear on the outside of EVs, no consensus exists on which nucleic acid species inside small EVs (<200 nm, sEVs) are sufficiently abundant and accessible for developing genotyping protocols. We investigated this by extracting total intravesicular nucleic acid content from sEVs isolated from the conditioned cell medium of the human NCI-H1975 cell line containing the epidermal growth factor (*EGFR*) gene mutation T790M as a model system for non-small cell lung cancer. We observed that mainly short genomic DNA (<35–100 bp) present in the sEVs served as a template. Using qEV size exclusion chromatography (SEC), significantly lower yield and higher purity of isolated sEV fractions were obtained as compared to exoEasy membrane affinity purification and ultracentrifugation. Nevertheless, we detected the *EGFR* T790M mutation in the sEVs’ lumen with similar sensitivity using digital PCR. When applying SEC-based sEV separation prior to cell-free DNA extraction on spiked human plasma samples, we found significantly higher mutant allele frequencies as compared to standard cell-free DNA extraction, which in part was due to co-purification of circulating tumor DNA. We conclude that intravesicular genomic DNA can be exploited next to ctDNA to enhance *EGFR* T790M mutation detection sensitivity by adding a fast and easy-to-use sEV separation method, such as SEC, upstream of standard clinical cell-free DNA workflows.

## 1. Introduction

Extracellular vesicles (EVs) form interesting diagnostic biomarkers for cancer genotyping in liquid biopsies. They are small cell-derived structures surrounded by a lipid bilayer that are released in the bloodstream. By transferring their cargo such as proteins, sugars, and genetic material to recipient cells, they influence various physiological, but also pathological functions [1]. For instance, EVs play a role in growth, development and blood coagulating, but also tumorigenesis by pre-metastatic niche formation and immune regulation [2,3]. Different types are classified based on their size, ranging from 30 to 5000 nm, and mechanism of biogenesis. We adopt below the definition of “small EV (sEV)” for EVs < 200 nm and “medium and large EVs” for those > 200 nm in line with the MISEV guidelines [1]. Mainly the sEVs are considered to assist in the early stages of cancer development and the interaction network of therapy resistance [2,3,4]. Importantly, the molecular cargo of EVs molecular cargo of EVs represent the cell from which they originate. Their nucleic acid content mainly consists of short non-coding RNAs (ncRNA) [5], but also messenger RNA (mRNA) molecules that can carry genomic gene mutations [6,7]. Moreover, Kahlert et al. (2014) and Thakur et al. (2014) reported DNA containing somatic gene mutations inside EVs [8,9]. Lázaro-Ibáñez et al. (2019) had similar results, and in addition found the highest coverage of human genes in DNase-treated EVs using whole genome sequencing [10]. Although not yet implemented in clinical laboratories, research studies have shown that RNA and/or DNA associated with EVs can improve the sensitivity of mutation detection in liquid biopsies from lung cancer patients [11,12,13,14]. Moreover, although most of the DNA has been found to be present on the outside of EVs, the coverage of human genes was higher in DNase-treated samples suggesting this genetic information is mainly transported as a luminal EV cargo [10]. However, it is not yet clear which is the predominant nucleic acid species that serves as a template for mutation detection inside EVs.

The future implementation of an EV-based assay in a molecular diagnostic setting requires a standardized, quick, and easy-to-use workflow of EV separation combined with a nucleic acid extraction method targeting the template for mutation detection. Although numerous EV separation methods are available, they result in different subpopulations that vary in size and/or cargo [15]. Differential centrifugation followed by ultracentrifugation (UC) is the most widely applied method in research, but many different protocols exist [15]. Furthermore, the technique is time-consuming and low-throughput, and therefore not suitable for clinical laboratories [16]. Different commercial kits based on size selection, precipitation or affinity interaction are on the market claiming quick and reproducible separation of EVs [16]. Whereas multiple studies describe the effect of different EV separation methods on the purity, yield and functionality of EVs, as well as their protein and/or nucleic acid content [17,18,19], a systematic comparison of downstream genotyping starting from EV-RNA/DNA is currently lacking.

In this study, we aimed to investigate which intravesicular nucleic acid species are sufficiently abundant and accessible for oncogene mutation detection and assess state-of-the-art EV separation methods for their performance in a genotyping workflow. We used a human epithelial non-small cell lung cancer (NSCLC) cell line with the well-characterized heterozygous epidermal growth factor receptor (*EGFR*) gene mutation T790M as a standardized tumor cell model with tyrosine kinase inhibitor resistance for generating liquid medium samples containing mutant sEVs. The sEVs were separated using commercially available qEV size exclusion chromatography (SEC) columns and compared with sEV separates obtained by exoEasy (EE) membrane affinity or ultracentrifugation (UC). We systematically characterized both the sEVs (yield, purity, concentration, size, morphology, and identity) and their RNA/DNA cargo (yield, fragment length). Differential testing of mutation detection in intravesicular RNA versus DNA templates was performed by droplet digital PCR (ddPCR), using nucleic acid extracts of sEVs that were treated with a mix of RNase and DNase. We selected the most suitable sEV separation method, combined with a cell-free DNA (cfDNA) extraction method currently used in clinical settings to evaluate our findings in a plasma background. Healthy donor plasma samples were spiked with cell line-derived sEV separates, alone or in combination with short-mutated DNA fragments, to represent an early or late disease stage, respectively [20]. Here, the allele frequency was compared between samples extracted by the standard cfDNA extraction method or after sEV separation.

## 2. Results

### 2.1. Characterization of sEVs

In order to study the nucleic acid species encapsulated in sEVs that is most dominant and accessible as a template for mutation detection, the human NSCLC-derived NCI-H1975 (further called H1975) cell line containing the *EGFR* gene mutation c.2369C>T, p.T790M (hereafter called *EGFR* T790M) was used. We separated sEVs from the cell media using SEC since this workflow is well known to deliver relatively pure sEV fractions [21,22]. To investigate the effect of different purities of sEV fractions on the downstream mutation detection we also purified sEVs using EE, as well as UC which is considered the ‘golden standard’ method for sEV separation. The presence of H1975 cell line-derived sEVs obtained by each separation method was confirmed by in-depth characterization of the protein and particle concentrations, transmission electron microscopy (TEM) images, and protein markers.

The concentration and size of particles were measured by scatter-based nanoparticle tracking analysis (NTA, Figure 1a and Appendix A). The SEC-derived samples contained the lowest particle number concentration (1.7 ± 0.3 × 10^10^ particles/mL), followed by UC samples (4.7 ± 1.3 × 10^10^ particles/mL) and EE samples (2.6 ± 0.7 × 10^11^ particles/mL). Both SEC and UC fractions showed a modal particle diameter of about 150 nm. In contrast, EE-derived samples revealed a large portion of particles with sizes > 200 nm and the EE blank control showed a similar concentration (2.4 ± 0.06 × 10^11^ particles/mL) and profile to the EE-derived EV fraction. Therefore, the blank-corrected concentration of EE-derived samples (6.5 ± 3.2 × 10^10^ particles/mL) is shown in Figure 1a. No events were observed in the SEC and UC blank controls. As a side observation, using high-sensitivity flow cytometry, EE-derived EVs were observed to show a higher forward scatter (FWSC) in fluorescence intensity-FWSC plots compared to SEC and UC-derived EVs (Appendix A), corresponding to the larger EV size observed by NTA. The mean concentration of proteins present in the samples (Figure 1a) was found to be lowest in SEC fractions. This was 22 and 45 times (statistically significant, *p* ≤ 0.01) higher in UC and EE fractions, respectively. SEC showed the highest particle-to-protein ratio, which was statistically significant (*p* ≤ 0.05) compared to EE and UC (Figure 1a), indicating the most complete separation of sEVs from proteins and highest purity. Morphology analysis by TEM (Figure 1b and full images, Appendix A) showed that SEC and UC samples contained typically cup-shaped sEVs, whereas EE samples contained particles with a size similar to sEVs, often tending to form clusters, which were also observed in the EE blank (Appendix A). These particle clusters, containing a mixture of non-EV contaminants and sEVs (*vide infra*), can explain the larger size and higher size variability observed using NTA and high-sensitivity flow cytometry in EE, compared to SEC and UC fractions.

Analysis of the canonical sEV markers cluster of differentiation 81 (CD81) and heat shock protein 70 (Hsp70), as well as two non-sEV markers, p-Ribosomal Protein S6 (rpS6) and calnexin (CANX) by Western blot (Figure 1c and full images, Appendix A) [1] revealed that all EV preparations contained the sEV markers CD81 and Hsp70. For SEC and EE, the non-sEV markers CANX and rpS6 were absent, and all blank controls were clean. For UC, however, we observed binding signal of the anti-rpS6 antibody at a height of 37 kDa where multiple bands in the positive control represent different degrees of phosphorylation. The percentage of sEVs expressing the tetraspanins CD9, CD63, and CD81 was determined using high-sensitivity flow cytometry (Figure 1d). We observed a lower mean percentage of positive sEVs for all three markers in SEC fractions compared to EE and UC, which was statistically significant (*p* ≤ 0.05) compared to EE.

Overall, SEC delivered the purest fraction of sEVs although with a low yield compared to the other methods UC and EE.

### 2.2. sEV Encapsulated RNA and DNA

For nucleic acid extraction we used the exoRNeasy kit, which was designed to work downstream of EE affinity columns. To assess the intraluminal nucleic acid content of separated sEVs, the fractions were treated with a mixture of RNase A/T1 and DNase I with 10× reaction buffer containing MgCl_2_ (called RNase/DNase, Figure 2a) prior to sEV lysis and nucleic acid extraction. The concentration was verified to ensure degradation of all non-encapsulated RNA and DNA in the samples. Validation of the RNase/DNase treatment on total RNA extracts of lysed sEVs (Appendix A) showed the presence of both RNA and DNA in the untreated samples, whereas these were successfully removed by the nuclease treatment. Nanoparticle tracking analysis of fluorescently labelled sEVs (fluo-NTA) confirmed that the size distribution profile was unaltered after RNase/DNase treatment which indicated that the sEVs were left intact (Figure 2b). Using fluorescent quantification assays we found that the RNase/DNase treatment resulted in a decrease of mean total RNA and DNA (Figure 2c, mean ± SD). For SEC 35 ± 34% of total RNA and 46 ± 64% of DNA was removed, and therefore not intraluminal present. The largest decrease of total RNA and DNA was however observed for EE (82 ± 5%, *p* ≤ 0.001 and 43 ± 9%, *p* ≤ 0.01, resp.). UC fractions showed a 22 ± 21% decrease in RNA and a slight increase for DNA (18 ± 34%) after nuclease treatment due to the large variation between different sEV separations. Both before and after RNase/DNase treatment, EE resulted in the highest concentration of sEV-specific RNA and DNA. When calculating RNA/DNA ratios (mean ± SD) in total nucleic acid extracts (without nuclease treatment), RNA was 14 ± 10 times more abundant than DNA in SEC fractions. For EE and UC this was 11 ± 3 and 8 ± 2 times, respectively. The ratio (mean ± SD) of sEV-encapsulated RNA over DNA after nuclease (Figure 2d) was also the highest for SEC (8.8 ± 2.3), followed by EE (2.5 ± 0.5) and UC (3.6 ± 0.5).

To obtain a better understanding of the intravesicular nucleic acid species, their fragment lengths were analyzed (Figure 2e). In the RNA fragment length distribution profiles marker peaks at either 25 nucleotides (nt; total RNA) or 4 nt (small RNA), and in the DNA profiles at 35 and 10,380 base pairs (bp) indicated that the electrophoretic chip runs were successful. Small EV-derived RNA without nuclease treatment obtained by SEC and UC contained RNA fragments shorter than 200 nt, and peaks around 2000 nt and 4000 nt that indicated the presence of 18S and 28S ribosomal RNA (rRNA), respectively. After treatment with RNase/DNase, the rRNA peaks in the UC and SEC profiles disappeared, whereas an increased proportion of fragments < 500 nt was observed. For EE samples, the fragment length distribution profile was not altered. When zooming in on the small RNA fragments < 150 nt of EV samples with RNase/DNase treatment, sEVs separated by UC and SEC showed a distinct peak between 50 and 70 nt in addition to fragments shorter than 40 nt, indicating the presence of microRNAs (miRNA), and other small RNAs up to 150 nt. For EE, we mainly observed fragments in the miRNA region below 40 nt. When the same samples were analyzed using the High Sensitivity DNA kit, a similar fragment distribution profile of very short DNA fragments in the range of <35 bp to 100 bp was observed for the UC and SEC workflows, coinciding with the marker peak at 35 bp. The EE profile revealed the largest fraction of DNA fragments, most of which were just over 50 bp in length, in agreement with the RNA/DNA ratio (Figure 2d).

In conclusion, RNA was more abundantly present than DNA in sEV obtained with all separation methods but depending on the used separation method, a different fragment length distribution profile of sEV-RNA/DNA was obtained.

### 2.3. Genomic sEV-DNA Provides a Major Template for Mutation Detection

Based on the acquired insight of the nucleic acid species present inside NSCLC cell line-derived sEVs, we were interested to know which served as a template for *EGFR* T790M mutation detection by ddPCR. H1975 sEVs treated with RNase/DNase were used to distinguish template RNA from DNA. We either supplemented the sEV-derived RNA/DNA with reverse transcriptase (RT) for complementary DNA (cDNA) synthesis, or not. Similarly treated H1975 cellular RNA was used as control sample. Endpoint droplet fluorescence analysis (Figure 3a) in sEV-derived RNA/DNA extracts obtained by SEC confirmed the detection of wild-type and *EGFR* T790M mutated sequences both with and without prior cDNA synthesis. In contrast, no amplification occurred in the H1975 cell RNA sample without RT, excluding the possibility of primer and probe binding to RNA. Therefore, we concluded that both RNA and dsDNA species harboring wild-type and mutated sequences resided inside the purified sEVs.

To assess the abundance of the *EGFR* mutation in the two nucleic acid species we performed absolute quantification by ddPCR and normalized it to an equal amount of conditioned cell culture medium for all workflows. The mean copy number concentration (±SEM) of *EGFR* T790M sequences (Figure 3b) in RNA was 3.1 (±0.2) × 10^2^ copies/µL in SEC-derived sEV samples. This was 2.15 (±0.09) × 10^2^ copies/µL for EE, and UC samples contained the highest concentration with 13 (±4) × 10^2^ copies/µL. The concentration in case of SEC was statistically significantly (*p* ≤ 0.05) higher than the one of EE. The DNA sequences in SEC-derived sEV samples contained 6 (±1) × 10^2^ copies/µL of *EGFR* T790M, which was higher than RNA. A possible explanation for this is that the RT reaction was not entirely linear. The cDNA was produced before sample partitioning for ddPCR and thus a bias can be introduced if the cDNA molecules do not accurately represent the initial number of target RNA molecules [23]. This was also the case for the EE-derived sEV-DNA showing a mutant concentration of 3 (±2) × 10^2^ copies/µL, but not for UC with 8 (±2) × 10^2^ copies/µL. No significant differences (α = 0.05) in DNA mutant copy number concentration were found between SEC, EE, and UC-derived sEV samples, or between RNA and DNA within the respective sEV samples.

Since several NSCLC cell lines (NCI-H460, NCI-H520, and NCI-H1299) have been reported to possess RT activity, we investigated if the template DNA present in the H1975 sEVs was of genomic origin [24]. We performed electrophoretic analysis of PCR amplicons based on the exon-intron structure of the *EGFR* gene. Two PCR primer pairs encompassing the mutant locus were designed, one to amplify a fragment of 80 bp within one exon (intra-exon) and one with the forward primer binding to a sequence located in one exon and the reverse primer on the junction with the next exon (exon–exon junction), resulting in a fragment of 75 bp (Figure 3c). Amplification of DNA was performed on a nuclease-treated sEV-RNA/DNA sample obtained by SEC. In parallel, H1975 cellular genomic DNA (gDNA) and H1975 cellular cDNA were processed. The fragment length profiles of the amplicons (Figure 3d) show amplification of cellular gDNA by the intra-exon primer pair, resulting in a peak of 80 bp, but not by the exon–exon junction primer pair. Cellular cDNA was amplified by both primer pairs, resulting in peaks at 80 bp and 75 bp, while the NTC showed no amplification. Starting from sEV-RNA/DNA a fragment of 80 bp was formed by the intra-exon primer pair, but no amplification occurred by the exon–exon junction primer pair, indicating that the DNA fragments present in sEVs are gDNA fragments.

To surmise, the amount of mutated sequences was not significantly different when an RT synthesis step was performed on the sEV-RNA/DNA samples, indicating that mainly DNA served as a template. In addition, we found that this sEV-DNA was amplificated like cellular genomic DNA when using specific primer pairs.

### 2.4. Mutation Enrichment by sEV Separation and Co-Isolated ctDNA

Based on our findings in the H1975 cell line-derived sEVs, we postulated that SEC-based sEV separation, prior to cell-free DNA extraction and mutation analysis in plasma may enhance the sensitivity of a standard clinical diagnostic workflow. To test this, five human healthy donor-derived platelet-poor plasma (PPP) samples were spiked with H1975-derived sEVs, alone (sEV) or in combination with *EGFR* T790M containing 134-bp DNA fragments mimicking ctDNA (sEV + ctDNA), to represent an early or late disease stage, respectively. The spike-in concentrations were chosen to be physiologically relevant, the H1975-derived sEV concentration containing *EGFR* T790M (1 × 10^8^ vesicles/mL plasma) was chosen to be about 100 times lower than the total reported 9.3 × 10^9^ to 2.4 × 10^10^ vesicles/mL of healthy donor plasma since only a fraction of all sEVs in lung cancer patients are derived from tumor cells. The concentration of DNA fragments (300 copies/mL plasma) is within the reported range of *EGFR* mutation during progressive disease in plasma samples, mean 125.0 copies/mL (20.5–15,608.0) of 14 patients [21,22]. Cell-free DNA was extracted using the QIAamp Circulating Nucleic Acid kit (QIA), either directly from the spiked PPP samples or from the PPP-derived sEV fractions separated by SEC (SEC-QIA).

The total amount of extracted DNA for the five donors is shown in Figure 4b. The mean (N = 5, *n* = 1) was not significantly altered (α = 0.05) by the spiked-in sEVs or sEVs + ctDNA, but was systematically higher in all donors when first sEVs were separated. Analysis of the *EGFR* T790M mutation in PPP-derived samples was performed by ddPCR (for individual donor results, see Appendix A). The *EGFR* T790M allele frequency (Figure 4b) was statistically significantly higher for sEV samples (36 ± 3%, *p* ≤ 0.01) and sEV + ctDNA samples (51 ± 6%, *p* ≤ 0.01) obtained by SEC-QIA compared to QIA (20 ± 5% and 26 ± 5%, respectively). In fact, sEV + ctDNA samples processed by SEC-QIA resulted in a statistically significantly higher *EGFR* T790M allele frequency compared to sEV samples processed by both QIA and SEC-QIA (*p* ≤ 0.01), indicating that co-isolated ctDNA also contributed to the enrichment of *EGFR* T790M in the sEV-based extraction workflow.

## 3. Discussion

Several studies have demonstrated the potential of sEVs in molecular diagnostics [10,11,13,25,26], but up to now it is not clearly understood whether and which RNA or DNA species in purified sEV fractions, more specifically localized inside the sEV lumen constitute a template for mutation detection. Moreover, the effect of different sEV separation methods on the potential for sEV-based cancer genotyping has not yet been evaluated. Here, we used the H1975 cell line expressing heterozygous *EGFR* T790M as a simplified lung epithelial cell model representative of NSCLC. For sEV separation we used qEV SEC which is easy to use in a standardized way and known to yield pure fractions, followed by total RNA/DNA extraction using the exoRNeasy kit. We were able to detect the *EGFR* T790M single point mutation using ddPCR in both RNA and DNA species located inside sEVs, even though the DNA template was less abundantly present than RNA. Moreover, we confirmed that this DNA template was of genomic origin. Using spiked human plasma samples, we confirmed that a workflow based on SEC for sEV separation allows for co-isolation of ctDNA and can enhance mutation detection sensitivity.

For comparison with SEC, we also applied EE, another commercially available column-based method, and UC as the “golden standard” for sEV separation, which are known to each generate different sEV quality and quantity. We found that EE and UC both resulted in co-elution of proteins, and that the EE fractions in addition showed contamination with non-sEV particles that were presumably eluting from the purification column. Based on size distribution analysis by NTA and flow cytometry we observed consistently larger particles in EE fractions compared to SEC and UC, which could be either due to a larger EV size or formation of agglomerates [27]. The protein marker profiles in the three different sEV separations were similar, but the tetraspanins CD9, CD81, and CD63 were most abundantly present in EE fractions (statistically significantly higher than SEC (*p* ≤ 0.05), but not than UC). This can be expected based on the fact that larger EVs are more enriched in proteins and are more likely to pass the fluorescence detection threshold in the applied flow cytometry method.

Following our applied workflows we observed RNA to be more abundant than DNA in the total sEV fractions, which is in agreement with observations on sEVs obtained from low-density fractions by Lázaro-Ibáñez et al. (2019) [10]. Moreover, the majority of both RNA and DNA was localized at the outside or on the surface of the EVs, in line with previous studies [10,28,29,30]. We successfully removed the latter species by RNase/DNase treatment to enable extraction of only EV-enclosed nucleic acid species. This was corroborated by fragment analysis, showing a decrease in the total RNA/DNA concentration and disappearance of ribosomal RNA peaks, in line with previous reports of pure sEV-RNA profiles obtained by density gradient purification [10,31]. We found that for SEC 46% and for EE 43% of the DNA was extraluminal, which is in contrast with the study of Lázaro-Ibáñez et al. (2019) who observed that 97% of DNA from cell line EVs in low density and over 79.9% in high-density fractions were extraluminal. However, in this study human leukemic cell lines with different culture conditions were used in contrast to the NSCLC cell line that was used in our study, also another centrifugation protocol for EV separation, as well as DNA extraction and quantification method was applied that can lead to different amounts of extraluminal DNA. The DNase treatment was similar to our study [10].

When studying the intraluminal nucleic acid species into more detail, SEC resulted in 10 times less sEV-encapsulated RNA and DNA compared to EE, but similar quantities were extracted with UC. Fragment length distribution analysis of SEC-derived sEV-RNA revealed fragments in the miRNA region (<40 nt), fragments between 50–70 nt and other small RNAs up to 150 nt, which can be in part attributed to transfer RNA (tRNA), or fragmented RNA and long non-coding RNA (lncRNA). A similar result was found for UC, and this is in line with previous studies in which mainly miRNA, but also tRNA and fragments of rRNA, mRNA, and lncRNA, as well as small non-coding RNA species were detected by next-generation sequencing (NGS) in sEV preparations obtained by different workflows including UC, qEV SEC and EE [32,33,34]. For EE, however, mainly fragments in the miRNA region were found in our study. For both SEC and UC, only low amounts of short DNA fragments (<35 bp to 100 bp) were observed, whereas for EE these fragments were more abundantly present with the majority were over 50 bp in length. This is in contrast with previous findings of Thakur et al. (2014), Kahlert et al. (2014), Cai et al. (2013), and Lee et al. (2014) who reported, in addition to short DNA fragments of 100 bp to 180 bp, the extraction of EV-enclosed DNA fragments of 2.5 kb and larger from cell lines or plasma [4,8,9,35]. However, our findings match with those of the low density fraction of sEVs from the HMC-1 cell line reported by Lázaro-Ibáñez et al. (2019) [10].

We detected the *EGFR* T790M mutation inside the H1975 sEVs obtained by three different isolation methods, despite substantial differences in total sEV yield, purity, and concentration, as well as intravesicular nucleic acid fragment lengths. Lázaro-Ibáñez et al. (2019) also found the highest coverage of human genes in DNase-treated samples which indicates these are mainly transported as a luminal EV cargo. This prompted us to investigate the abundance and nature of the intravesicular templates for ddPCR. Although the exoRNeasy kit was designed to work downstream of EE affinity columns and the EE workflow resulted in the highest yield of both RNA and DNA, we found that SEC and UC delivered a higher copy number concentration of the mutation inside the sEVs. Using standard cDNA synthesis before absolute quantification by ddPCR, SEC showed a statistically significant (*p* ≤ 0.05) higher concentration compared to EE. Interestingly, we also detected the mutated sequences in all sEV samples without prior reverse transcription, indicating the presence of template DNA. Although the SEC workflow resulted in the highest intraluminal RNA:DNA ratio (9 ± 2), it yet rendered a higher concentration of mutation without the reverse transcriptase step (although not statistically significant). The same was observed for the EE, but not the UC workflow, in which case RNA templates seemed to be most abundant. This suggests the separation of sEVs with a large share of short, non-template or non-coding RNA fragments by SEC that do not contribute to the mutation detection. Using well-designed intra-exon and exon–exon junction PCR primer pairs, we furthermore confirmed that the template DNA enclosed in the H1975 sEVs was of genomic origin, and not cDNA. This is in accordance with the studies of Cai et al. (2013) and Kahlert et al. (2014) who detected genomic DNA inside EVs using NGS [8,35].

To translate our findings in cell line sEVs to a molecular diagnostic workflow, we applied SEC upstream of a cell-free DNA extraction method as currently used in clinical settings to human PPP samples spiked with H1975 cell line-derived sEVs, alone or in combination with short DNA fragments containing the *EGFR* T790M mutation (134 bp, mimicking ctDNA). When DNA was extracted after sEV separation a significantly higher *EGFR* T790M allele frequency (*p* ≤ 0.01) was observed for both types of spike-in samples. The highest *EGFR* T790M allele frequency occurred when both sEVs and ctDNA were spiked in the plasma samples, which shows that ctDNA is able to co-isolate with sEVs using the SEC method. This is in accordance with the study of Zocco et al. (2020) where they found higher copy numbers of melanoma mutant gene *BRAF* V600E when they applied peptide-based affinity EV isolation before DNA extraction in comparison to standard cell-free DNA extraction [36]. Recent studies have observed the formation of a protein corona on the surface of sEVs in plasma [37], which may imply that ctDNA in the blood will also associate with sEVs which may contribute to previous findings of DNA on the outside of sEVs [8,9,10].

We have proven here that a fast and easy-to-use column-based sEV separation provides sufficient sEV-enclosed template for *EGFR* T790M genotyping and that most of the mutation is located in genomic DNA, and not sEV-RNA species. Small EV-derived DNA can thus contribute to increasing the sensitivity of currently applied ctDNA analysis, by adding an sEV enrichment and lysis step to molecular diagnostic workflows, without the need for cDNA synthesis. Using spiked-in plasma samples we show that this method of sEV separation in combination with a cell-free DNA extraction method currently used in the clinic results in higher mutant allele frequencies compared to the standard cell-free DNA extraction method on.

It remains to be investigated whether SEC as sEV separation method provides sufficient templates for genotyping starting from patient plasma samples, since in previous studies so far mainly UC was used [8,9,13,14,38]. Castellanos-Rizaldos et al. (2018) found an increase of clinical sensitivity with 30% and of specificity with 9% when adding EV-derived nucleic acids to ctDNA for earlier disease stages [12]. Wan et al. (2018) had a similar conclusion, but found in addition that the increased sensitivity vanished in late-stage NSCLC [13]. Possibly additional sensitivity can be obtained by enriching tumor-related sEVs using affinity-based separation with magnetic beads or flow cytometry sorting. So far, allele-specific PCR has been used in genotyping studies, however, there is increasing clinical interest in multiplex analysis of mutations in multiple genes by NGS. Since the sensitivity necessary for early diagnostic population screening using NGS profiling on ctDNA remains challenging [39,40], here too, the clinical implementation of a combined sEV-DNA and ctDNA approach can provide a major advancement.

## 4. Materials and Methods

### 4.1. Cell Culture

Human epithelial NSCLC cell line NCI-H1975, which is heterozygous for the *EGFR* T790M mutation, was maintained in RPMI-1640 medium with GlutaMAX (Gibco, Thermo Fisher Scientific, Grand Island, NY, USA, #72400021) supplemented with 10% FBS (Gibco, Thermo Fisher Scientific, #A5256701). During culture, cells were incubated at 37 °C in 5% CO_2_ and 95% relative humidity. Every 3–4 days, 1.1 × 10^4^–1.3 × 10^4^ cells/cm^2^ were passaged to new T75 cm^2^ tissue culture treated cell culture flask. During maintenance, the cell cultures were regularly tested for absence of mycoplasma contamination using the MycoAlert Mycoplasma Detection Kit (Lonza, Basel, Switzerland, #LT27-236).

### 4.2. Preparation of Conditioned Cell Media and Blank Controls

For EV production, 8.5 × 10^5^ cells were seeded in a T75 cm^2^ flask in 20 mL of their respective culture media supplemented with 10% FBS and grown for 3 days until they reached about 70% confluency. Then, cells were washed two times with 5 mL PBS before changing to 15 mL cell culture medium supplemented with 2% EV-depleted FBS (Gibco, Thermo Fisher Scientific, #A272080) and incubated for another 24 h. The conditioned cell culture media were collected and centrifuged at 4 °C, 150 rcf for 10 min in a swinging bucket Rotanta 460 R centrifuge. Cell viability was assessed in parallel using a NucleoCounter NC-100 and assured to be >95%. For UC and SEC, the conditioned cell media were further purified by differential centrifugation steps to remove cellular debris: 300 rcf for 10 min, 500 rcf for 10 min, and 4500 rcf for 30 min at 4 °C in a swinging bucket Rotanta 460 R centrifuge. For EE, the medium was further cleared from large-sized contaminants by filtration over a 0.8 µm and a 0.22 µm pore size filter (Merck, Darmstadt, Germany, #AAWP02500 and #SLGV004SL). Blank controls consisted of unconditioned media supplemented with 2% EV-depleted FBS (RPMI-1640 + 2% EV-depleted FBS) that were processed in exactly the same way as the respective conditioned cell culture media.

### 4.3. Preparation of sEV Spiked Platelet-Poor Plasma

Five healthy donors were included in this study which was approved by the Ethics committee of Antwerp University Hospital, Antwerp, Belgium under code 2021-0399. Whole blood was collected in RNA complete blood collection tubes (Streck, La Vista, NE, USA, #230580) by venipuncture and a 21G needle. The tubes were immediately mixed 10 times by gentle inversion and stored upright at room temperature. Platelet-poor plasma (PPP) was prepared < 2 h after blood collection in accordance with the recommendation of the International Society on Thrombosis and Haemostasis as described by Lacroix et al. (2012) [41]. The whole blood was centrifuged for 15 min at 2500 rcf, the plasma was collected up to 1 cm above the buffy coat and transferred to a new 15 mL centrifugation tube. The plasma was centrifuged for 15 min at 2500 rcf and transferred to a new tube leaving about 100 µL to avoid transfer of platelets. After filtration through a 0.8 µm pore size filter to remove any remaining platelets the PPP was obtained. This was aliquoted, and either left untreated or spiked with sEVs obtained by UC (1 × 10^8^ sEVs/mL plasma), or sEVs in combination with 300 copies/mL of 134-bp *EGFR* T790M DNA fragments (5′-AGC GTG GAC AAC CCC CAC GTG TGC CGC CTG CTG GGC ATC TGC CTC ACC GTG CAG CTC ATC ATG CAG CTC ATG CCC TTC GGC TGC CTG GAC TAT GTC CGG GAA CAC AAA GAC AAT ATT GGC TCC CA-3′, gBlocks Gene Fragments, IDT, Leuven, Belgium) and stored at −80 °C until further use. PPP was thawed at room temperature and centrifuged for 15 min at 2500 rcf before use.

### 4.4. Size Exclusion Chromatography

The purified conditioned cell media of six T75 flasks and blank controls (90 mL) were concentrated to a volume of maximum 500 µL using a 100 kDa nominal molecular weight limit Centricon Plus-70 Centrifugal Filter Unit (Sigma-Aldrich, Saint Louis, MO, USA, #UFC710008) according to the manufacturer’s protocol. The concentrated conditioned cell medium or 500 µL of PPP were loaded onto a qEVoriginal 70 nm column (Izon Science Ltd., Lyon, France, #SP1), and immediately fractions of 500 µL were collected with PBS as elution buffer. Fractions 1 to 4 after the void volume were pooled (2 mL). Cell-line derived fractions were transferred to a 5 mL open-top thinwall polyallomer tube and centrifuged at 4 °C; 100,000 rcf for 65 min using an Optima XPN-80 ultracentrifuge equipped with a swinging bucket SW55 Ti rotor. The pellet was resuspended in 500 µL of PBS. PPP-derived fractions were immediately used for DNA extraction (see Section 4.8).

### 4.5. Exoeasy Membrane Affinity Chromatography

The purified conditioned cell media of three T75 flasks and blank controls (45 mL) were pooled and processed over one spin column of the exoEasy Maxi Kit (Qiagen, Hilden, Germany, #76064) according to the manufacturer’s protocol. Here 45 mL was used as start volume for the sEV separation method to avoid saturation of the affinity column. For details, see Appendix A. Small EVs were eluted in 500 µL.

### 4.6. Ultracentrifugation

The purified conditioned cell media of six T75 flasks were pooled (90 mL) or the same volume of blank control was used and divided over thinwall polypropylene tubes in six fractions of 15 mL. To remove larger particles these fractions were centrifuged at 4 °C; 10,000 rcf for 30 min in an Optima XPN-80 ultracentrifuge equipped with a swinging bucket SW32.1 Ti rotor, resulting in a clearing factor (k-factor) of 3797. The pellets were discarded, and 14.5 mL of the supernatants were transferred to new tubes by slowly pipetting from the top and centrifuged at 4 °C; 100,000 rcf (k-factor: 380) for 65 min. All pellets were resuspended in a total volume of 500 µL PBS.

### 4.7. Characterization of sEV Separations

For more details on the characterization methods below, see Appendix A. Protein concentration determination in sEV- or blank control samples was done using the DC Protein Assay (Bio-Rad Laboratories, Hercules, CA, USA, #5000112) for UC and EE, and the Micro BCA™ Protein Assay Reagent Kit (Thermo Fisher Scientific, #23235) for SEC according to the manufacturers’ guidelines.

Scatter-based NTA was performed on a NanoSight NS500 instrument. Each measurement consisted of three recordings of 60 s (1500 frames with 25 frames/second) using the following settings: camera level 14–16, screen gain 1.0. Detection threshold 5 was used for analysis. For fluorescent NTA, the sEVs separated by SEC were labelled with the membrane intercalating dye cell mask green (CMG, ThermoFisher Scientific, #C37608) before and after RNase/DNase treatment. CMG was 100 times diluted before adding 2 µL with 8 µL sEVs and incubation at room temperature for 1 h in the dark. The mixture was lengthened to 1000 µL with milli-Q water before fluo-NTA analysis with a ZetaView TWIN (Particle Metrix) equipped with a 488 nm laser. Each sample was measured 3 times at 12 positions with low bleach function and shutter set at 100 and sensitivity at 97.

TEM images were taken using a Tecnai G2 Spirit BioTWIN TEM instrument in Bright Field modus at 120 kV. Western blotting was done using primary antibodies (all purchased from Santa Cruz Biotechnology, Dallas, TX, US) for detection of CD81 (1/200, 200 µg/mL, #sc-166028), Hsp70 (1/500, 200 µg/mL, #sc-24), CANX (1/500, 200 µg/mL, #sc-23954), and rpS6 (1/200, 200 µg/mL, #sc-293144). High-sensitivity flow cytometry analysis was done after staining sEV samples with CFDA-SE (Thermo Fisher Scientific, #V12883) and 0.5 µg of a phycoerythrin-Cy7 tandem conjugate (PE-Cy7) fluorescently labelled mouse anti-human antibody (BioLegend) against: CD9 (50 µg/mL, #312115), CD63 (200 µg/mL, #353009), or CD81 (200 µg/mL, #349512) based on the workflow described by Deville et al. (2021) [27]. Samples were purified by bottom-up iodixanol density gradient centrifugation to remove the unbound dye and analyzed using a BD Influx flow cytometer equipped with a 488 nm high power laser (200 mW) and a small-particle detector as previously described by van der Vlist et al. (2012) [42].

### 4.8. RNA/DNA Extraction

The total nucleic acid content was extracted from the cell line-derived sEV fractions using the exoRNeasy Maxi kit (Qiagen, #77164) according to the manufacturer’s protocol, consisting of sEV lysis followed by total RNA extraction and DNA co-extraction. To enable the extraction of nucleic acid species exclusively associated with the sEVs, sEV samples were treated with ribonuclease (RNase) and deoxyribonuclease (DNase) to deplete non-encapsulated RNA and DNA prior to vesicle lysis with the QIAzol Lysis Reagent. The used protocol was adapted from Enderle et al. (2015) using a mixture of 5 µL RNase A/T1 (2 mg/mL RNase A; 5000 U/mL RNase T1), 1 µL DNase I (1 U/µL), and 50 µL 10× reaction buffer with MgCl_2_ [29]. For more information see Appendix A. The removal of all non-encapsulated RNA and DNA by this treatment was validated (Appendix A). The RNase/DNase mixture was inactivated by addition of 2.5 mL QIAzol Lysis Reagent, followed by a 5-min incubation and addition of 500 µL chloroform and incubation for 3 min. All following steps were performed according to the manufacturer’s instructions. For preparation of genomic DNA and cellular RNA, see Appendix A.

PPP samples (0.5 mL) and pooled PPP-derived sEV fractions (2 mL) were processed using the QIAamp Circulating Nucleic Acid Kit (Qiagen, #55114) according to the manufacturer’s protocol. PPP samples were diluted to 1 mL using PBS and the PPP-derived sEV separations were supplemented with 0.01 g/mL bovine serum albumin (Sigma-Aldrich, Merck, Saint Louis, MO, USA, #A3059) to ensure complete DNA extraction. Proteinase K and lysis buffer ACL containing 1.0 µg carrier RNA were mixed with the samples in the prescribed volumes. The mixture was incubated at 60 °C for 30 min before adding buffer ACB and incubating on ice for 5 min. The lysate was passed through the QIAamp Mini column by vacuum force and washed with 600 µL buffer ACW1, 750 µL buffer ACW2 and 750 µL ethanol. The QIAamp Mini column was dried by centrifugation at 20,000 rcf for 3 min using a microcentrifuge 5415 R (Eppendorf) and subsequent incubation at 56 °C for 10 min with an open lid. DNA was eluted in 20 µL buffer AVE.

### 4.9. Small EV-RNA/DNA Yield and Purity Assessment

The Qubit RNA HS and dsDNA HS Assay Kits (Thermo Fisher Scientific, #Q32852 and # Q32854) were used to determine RNA (quantitative range of 250 pg/µL to 100 ng/µL) and DNA (10 pg/µL to 100 ng/µL) concentrations, respectively. Analysis was done using a Qubit 2.0 Fluorometer. Each sample was diluted with RT-PCR grade water (Invitrogen, Thermo Fisher Scientific, Waltham, MA, USA, #AM9935) to match the assay range and 1–20 µL was added to 199–180 µL of Qubit working solution. Before each use, new standards were used for calibration. Length profiling of sEV-RNA/DNA fragments was done using the Agilent RNA 6000 Pico Assay (25 nt to 6000 nt, Agilent Technologies, Santa Clara, CA, USA, #5067-1513), Agilent small RNA Assay (6 nt to 150 nt, #5067-1548) and Agilent High Sensitivity DNA Assay (50 bp to 7000 bp, #5067-4626). Samples were diluted with RT-PCR grade water to match the assay’s qualitative range and analyzed on a Bioanalyzer 2100 with the 2100 Expert Software, version B.02.10.SI764.

### 4.10. Mutation Detection

To distinguish between sEV-derived RNA and DNA as template for genotyping (see below), the extracted nucleic acids were either reverse transcribed into cDNA or left without reverse transcriptase (RT) treatment. For cDNA synthesis with the Transcriptor First Strand cDNA Synthesis kit (Hoffmann-La Roche, Basel, Switzerland, #4896866001), 10 µL total nucleic acids from sEVs treated with RNase/DNase prior to lysis, containing 18–74 ng for UC, 16–49 ng for SEC, and 280–508 ng for EE, or 1000 ng cellular RNA were taken per reaction. Anchored oligo (dT)18 and random hexamer primers were used, and the reaction was done in a Veriti 96-Well Thermal Cycler according to the manufacturer’s protocol.

Mutation detection was done using the primer-probe mix of the validated *EGFR* p.T790M detection assay (Bio-Rad Laboratories, #dHsaMDV2010019) containing probes with different fluorescent dyes to target the wild-type (hexachloro-carbonyl-fluorescein, HEX) and T790M (6-carboxy fluorescein, FAM) *EGFR* sequences. For more details on the methods below, see Appendix A. For ddPCR, a QX200 Droplet Digital PCR System was used. Results were analyzed using QuantaSoft Software, version 1.7.0. Only wells with a droplet count ≥ 10,000 were included for analysis. For H1975 cell-line derived samples, the threshold for fluorescence channel 1 (detection of FAM) was set at 2500 and the threshold for fluorescence channel 2 (detection of HEX) was set at 1700. All experiments were performed in technical triplicates. Results were normalized to the same starting volume for each of the separation methods (90 mL conditioned media). For PPP-derived samples the fluorescence thresholds were set at 2000 for both channels, all samples were measured in technical duplicated. The allele frequency was calculated by dividing the allele occurrence by the total copy number of the gene.

### 4.11. Confirmation of Genomic DNA Present in sEVs

SEC-derived H1975 sEVs were used as templates for PCR. Different primer pairs were designed to amplify a short fragment of the *EGFR* gene using the OligoAnalyzer Tool (Integrated DNA Technologies) and BLAST alignment (National Center for Biotechnology Information) to ensure the selectivity of the selected primers (Table 1). The first primer pair was designed to amplify a fragment within one exon (intra-exon, 80 bp). For the second pair, the forward primer was chosen to bind in one exon whereas the reverse primer was designed to bind on the junction of this exon with the next exon (exon–exon junction, 75 bp). PCR was performed using the 2× Platinum™ II Hot-Start PCR Master Mix (Invitrogen, #14000013). The obtained PCR products were purified using the DNA Clean & Concentrator Kit (Zymo Research, Irvine, CA, USA, #ZY-D4013) according to the manufacturer’s protocol. PCR products were analyzed using the Agilent High Sensitivity DNA Assay and a Bioanalyzer 2100 with the 2100 Expert Software, version B.02.10.SI764.

### 4.12. EV-TRACK, ExoCarta and Vesiclepedia

We have submitted all relevant data of our experiments to the EV-TRACK knowledgebase (EV-TRACK ID: EV200110) [15]. All proteins and mRNA sequences located inside sEVs detected in this study were added to both ExoCarta and Vesiclepedia web-based databases [43,44].

### 4.13. Statistical Analysis

Independent experiments are indicated with N, whereas technical replicates are noted as n. For studies with N > 1 and *n* = 1, or N = 1 and *n* > 1, data are represented as mean ± standard deviation (SD). For studies with N > 1 and *n* > 1, data are represented as mean ± standard error of the mean (SEM). Statistical analyses were done using a paired sample t-test (measurements performed on dependent samples), Student’s t-test (samples with equal variances) or Welch’s t-test (samples with unequal variances) and α = 0.05.

## 5. Conclusions

In conclusion, using a SEC column-based sEV separation method we showed that most of the *EGFR* T790M mutation is present in intraluminal short (<35–100 bp) DNA of genomic origin. This source of sEV-DNA can be readily detected by ddPCR, and also when considering other sEV separation methods based on EE or UC. A clinically translated workflow consisting of SEC for fast and easy sEV separation, followed by cell-free DNA extraction and mutation analysis, when applied to spiked plasma samples resulted in the detection of higher mutant allele frequencies. Therefore, we suggest that this may be a promising approach for future clinical diagnostics.

## Figures and Tables

**Figure 1 ijms-23-16052-f001:**
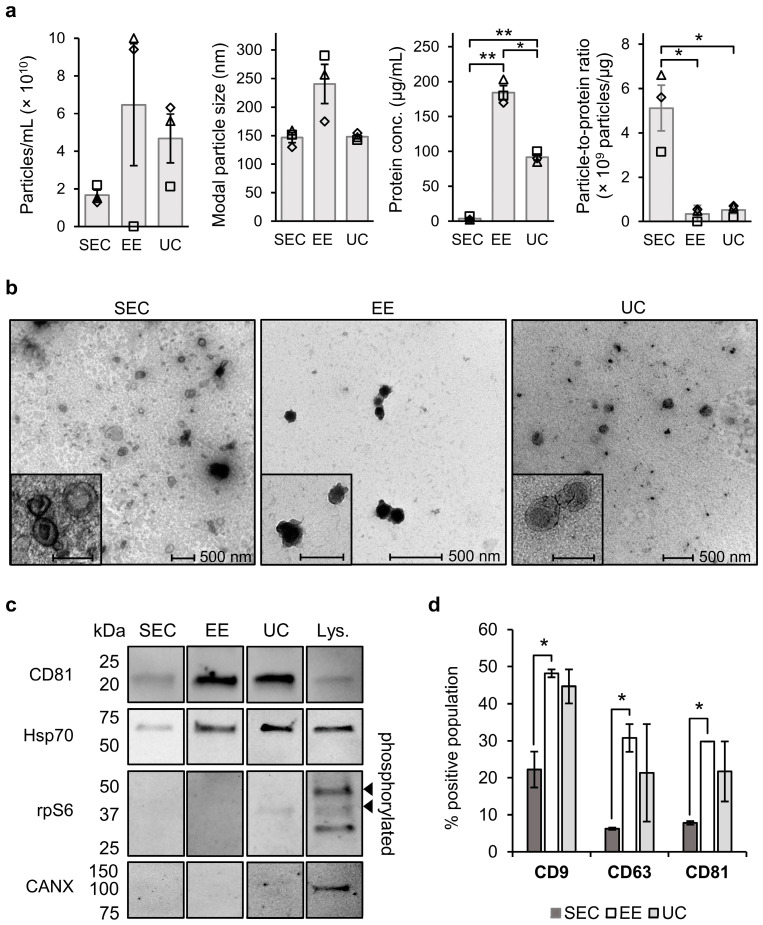
Characterization of H1975 small extracellular vesicles (sEVs) purified by size exclusion chromatography (SEC), in comparison with exoEasy (EE) and ultracentrifugation (UC). (**a**) Particle and protein analysis in sEV fractions isolated from conditioned cell media using SEC, EE, and UC. Particle concentrations (blank-corrected) and modal particle diameter were determined using scatter-based nanoparticle tracking analysis (NTA) using a NanoSight NS500, protein concentrations were measured using the Micro BCA™ Protein Assay Reagent Kit (SEC) and DC Protein Assay Kit (UC, EE). The results are presented as the mean ± SEM (N = 3, *n* = 3) or mean ± SD for SEC (N = 3, *n* = 1) of N experiments, each indicated by an open symbol. The blank-corrected particle-to-protein ratio was calculated as a measure of sEV purity. (**b**) Transmission electron microscopy images of sEVs represented with a scale bar of 500 nm (200 nm in insets). (**c**) Representative western blot analysis of sEV-specific protein markers (CD81, MW = 24–26 kDa and Hsp70, MW = 70 kDa) and non-sEV markers (rpS6, MW = 34 kDa and CANX, MW = 90 kDa) in sEV samples and whole H1975 cell lysate (Lys.). (**d**) Flow cytometry analysis of the percentage double positive population of CFDA-SE/CD9 (CD9), CFDA-SE/CD63 (CD63) and CFDA-SE/CD81 (CD81) versus CFDA-SE single stained sEV samples obtained using a BD Influx flow cytometer with a small-particle detector. Data is represented as mean ± SEM (N = 2, *n* = 3). Statistically significant results are indicated by * for *p*-value ≤ 0.05 and ** for *p*-value ≤ 0.01.

**Figure 2 ijms-23-16052-f002:**
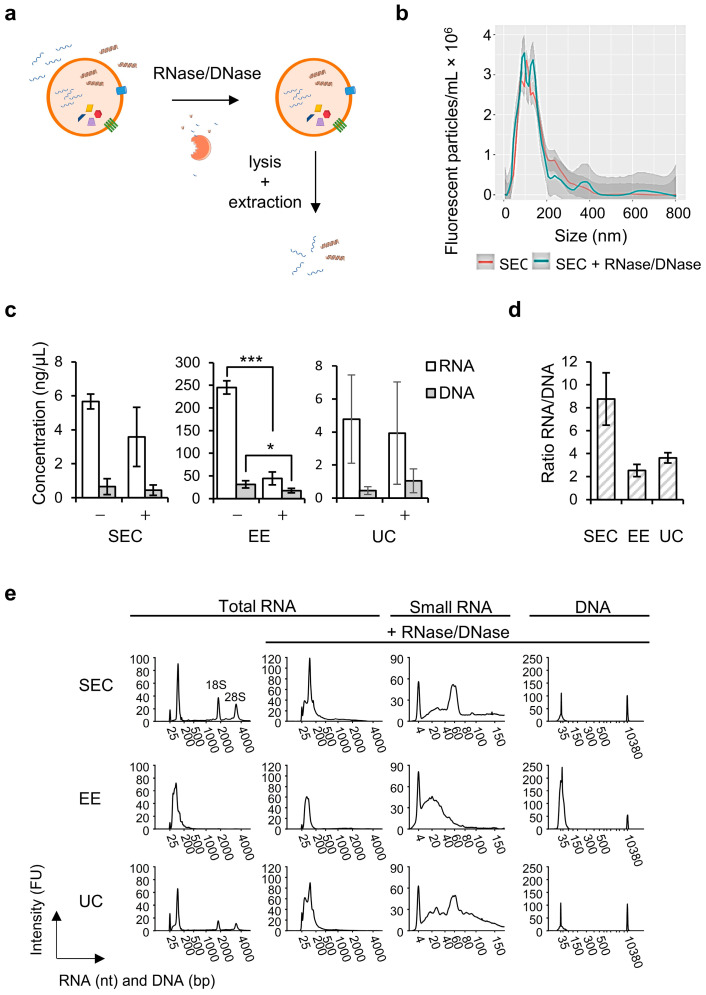
Characterization of intravesicular RNA and DNA species. (**a**) Schematic overview of DNase/RNase treatment before sEV lysis and sEV-contained nucleic acid extraction. (**b**) Nuclease treatment removes non-sEV encapsulated RNA and DNA, but leaves the sEV-membrane intact. Size distribution profiles of sEV samples separated by SEC that were either left untreated or treated with RNase/DNase, obtained by fluorescence NTA measured using a ZetaView TWIN. Samples were stained using the membrane intercalating dye cell mask green. The graph is shown as mean ± SD (*n* = 12) and is a representative of 3 independent measurements. (**c**) Concentration of total RNA and DNA in samples without treatment (-) and with RNase/DNase treatment (+) measured using the RNA HS and DNA HS assay, respectively, on a Qubit 2.0 Fluorometer. Data are represented as mean ± SD (N = 3). Statistically significant results after treatment with RNase/DNase are indicated by * for *p*-value ≤ 0.05 and *** for *p*-value ≤ 0.001. (**d**) Ratio of RNA over DNA with RNase/DNase treatment calculated from Figure 1c. (**e**) Fragment length distribution profiles of samples without treatment and with RNase/DNase treatment obtained using a Bioanalyzer 2100 and RNA 6000 Pico kit (total RNA), small RNA kit (small RNA) or High Sensitivity DNA kit (DNA). Representative profiles of three biological replicate experiments (N = 3) are shown. FU, fluorescence units; nt, nucleotides; bp, base pairs.

**Figure 3 ijms-23-16052-f003:**
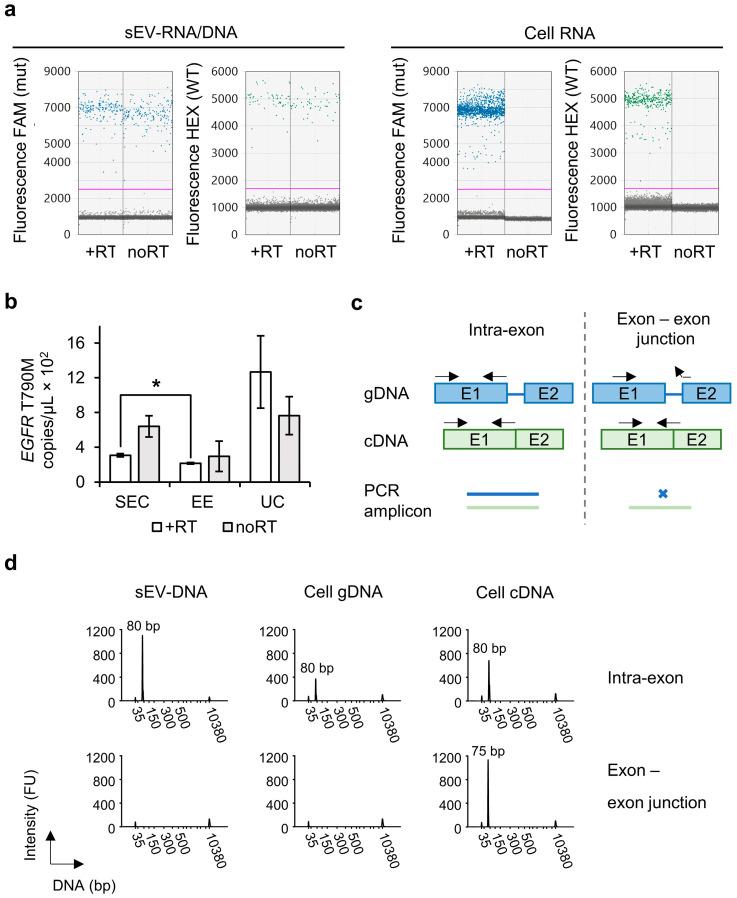
H1975 cell line sEV-derived genomic DNA provides a major template for *EGFR* T790M mutation detection. (**a**) Fluorescence intensities of mutant (FAM, mut) and wild-type (HEX, WT) obtained by a X200 Droplet Digital PCR System. Data are shown as 1D droplet amplitude with the threshold (pink line) separating positive from negative droplets. The sEV-derived RNA/DNA obtained by SEC and cellular RNA samples were used for cDNA synthesis using reverse transcriptase (+RT) or left untreated (noRT). (**b**) Concentration of *EGFR* T790M (copies/µL) in sEV-RNA/DNA obtained by ddPCR. Samples were prepared with reverse transcriptase step for cDNA synthesis (+RT, white) or left untreated (noRT, grey). Data obtained by SEC are compared to those of EE and UC methods. They are normalized to the same volume of conditioned culture medium used as starting material (90 mL) and represented as mean ± SEM (N = 3, *n* = 3). Statistically significant results are indicated by * for *p*-value ≤ 0.05. (**c**) The impact of PCR primer pair design on amplicon generation from gDNA (blue) and cDNA (green) containing both exons (boxes) and introns (lines) or only exons, respectively. Left: The intra-exon primer pair generates amplicons of the same length from both gDNA (blue) and cDNA (green). Right: The exon–exon junction primer pair generates an amplicon from cDNA, but not from gDNA. (**d**) Fragment length distribution profiles of PCR amplicons generated from sEV-RNA/DNA samples obtained by SEC, compared to cell-derived gDNA and cDNA. Intra-exon (top) and exon–exon junction (bottom) primer pairs were used for PCR. Analysis was done using a Bioanalyzer 2100 combined with the High Sensitivity DNA kit showing marker peaks at 35 bp and 10,380 bp. Small EV-derived samples were run undiluted and cellular samples were run after a 5-fold dilution. FU, fluorescence units; bp, base pairs.

**Figure 4 ijms-23-16052-f004:**
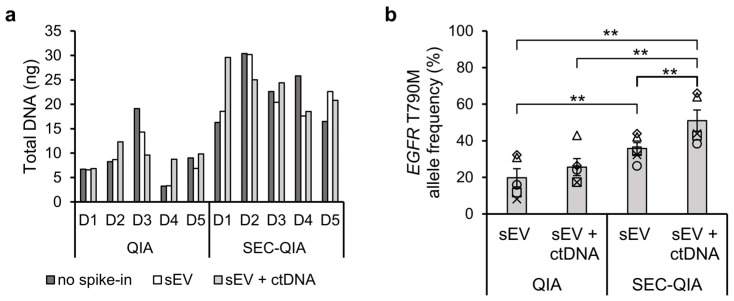
*EGFR* T790M enrichment in cell-free DNA extracts of human plasma extracted with the QIAamp Circulating Nucleic Acid kit (QIA), directly, or after separation of sEV fractions by SEC (SEC-QIA). (**a**) Concentration of total DNA in plasma samples from 5 healthy donors (D1-5) without spike-in (no spike-in), spiked with 1 × 10^8^ sEVs/mL plasma (sEV), or with sEVs in combination with 300 copies short *EGFR* T790M DNA fragments (sEV + ctDNA) using the HS DNA assay on a Qubit 2.0 Fluorometer. (**b**) *EGFR* T790M allele frequency in sEV and sEV + ctDNA samples obtained by ddPCR. Data are represented as mean ± SEM (N = 5, *n* = 2) of N donors, each indicated by an open symbol. Statistically significant results are indicated by ** for *p*-value ≤ 0.01.

**Table 1 ijms-23-16052-t001:** Sequences of primers used in PCR. FWD: forward primer, REV: reverse primer, bp: base pairs.

Primer Pair		Sequence 5′ → 3′	Amplicon Length (bp)
Exon–exon	FWD:	ATCTGCCTCACCTCCAC	80
	REV:	TTGTGTTCCCGGACATAGTC	
Exon–exon junction	FWD:	CAATATTGGCTCCCAGTACCTG	75
	REV:	CGACGGTCCTCCAAGTAGTTC	

## Data Availability

Not applicable.

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
