# Peer review of "Intravesicular Genomic DNA Enriched by Size Exclusion Chromatography Can Enhance Lung Cancer Oncogene Mutation Detection Sensitivity"

_ijms, 2022, doi:10.3390/ijms232416052_

Round 1

Reviewer 1 Report

The manuscript of Hoof et al., entitled “Intravesicular genomic DNA enriched by size exclusion chromatography can enhance lung cancer oncogene mutation detection sensitivity” investigated the intravesicular nucleic acid species that are sufficiently abundant and accessible for oncogene mutation detection. They found that mainly short genomic DNA present in the sEVs served as a template. Finally, they concluded that the sEVs genomic DNA can be exploited with ctDNA to enhance EGFR T790M mutation detection sensitivity by adding the sEV separation method. This is a very well-articulated article. However, some concerns needed to be addressed before publication. Please find below the minor concerns of this study.

Comments:

1.     In Figure 1 B, the size of sEVs in the EE group is comparatively larger than  SEC and UC groups which are not in line with the size shown by TEM.

2.     In Figure 1 A, is that mode of the sEVs?

3.     As mentioned in Supplementary Figure S1, the EE group showed significant sEVs in EE blank group. Then how the particle-to-protein ratio was calculated. Whether the subtraction of the blank was considered during the calculation.

4.     The authors have represented the Flow data in % positive events. They are suggested to represent figure 1 d  in % positive population instead of % positive events.

Reviewer 2 Report

This is a well performed study that evaluated several extracellular-vesicle enrichment/isolation strategies for detection of intravesicular nucleic acid sequences.  The experiments are clearly described with extensive documentation in the methods, and the results are appropriately characterized (and fully available between the paper and the supplemental information).  The conclusions are appropriate with the results presented, and the limitations and next steps of the work are defined.  The introduction places the work into appropriate context.  Overall, this is a high quality manuscript.
